# An Efficient Image Cryptosystem Utilizing Difference Matrix and Genetic Algorithm

**DOI:** 10.3390/e26050351

**Published:** 2024-04-23

**Authors:** Honglian Shen, Xiuling Shan

**Affiliations:** 1School of Mathematical Sciences, Hebei Normal University, Shijiazhuang 050024, China; shl218@126.com; 2Department of Mathematics and Computer Science, Hengshui University, Hengshui 053000, China

**Keywords:** image cryptosystem, difference matrix, genetic algorithm, optimal preservation strategy

## Abstract

Aiming at addressing the security and efficiency challenges during image transmission, an efficient image cryptosystem utilizing difference matrix and genetic algorithm is proposed in this paper. A difference matrix is a typical combinatorial structure that exhibits properties of discretization and approximate uniformity. It can serve as a pseudo-random sequence, offering various scrambling techniques while occupying a small storage space. The genetic algorithm generates multiple ciphertext images with strong randomness through local crossover and mutation operations, then obtains high-quality ciphertext images through multiple iterations using the optimal preservation strategy. The whole encryption process is divided into three stages: first, the difference matrix is generated; second, it is utilized for initial encryption to ensure that the resulting ciphertext image has relatively good initial randomness; finally, multiple rounds of local genetic operations are used to optimize the output. The proposed cryptosystem is demonstrated to be effective and robust through simulation experiments and statistical analyses, highlighting its superiority over other existing algorithms.

## 1. Introduction

With the increasing transmission of multimedia information over the internet, it has become essential to protect against unauthorized access. Digital images are a widely used data format for storing more vivid information. Image encryption mainly involves converting a meaningful image into a meaningless one through two fundamental techniques, namely, scrambling and diffusion, thereby rendering it inaccessible to unauthorized individuals. Scrambling involves the modification of the positioning or ordering of pixels within an image. On the other hand, diffusion modifies pixel values to achieve a randomized effect. These techniques can be applied independently or combined to create more complex encryption schemes.

Numerous image encryption algorithms have been put forward recently, with chaos-based ones garnering significant attention in this field due to their distinctive properties [1,2,3]. Chaotic systems are nonlinear dynamical systems that exhibit complex and unpredictable behavior, making them suitable for cryptographic systems due to their ergodicity, sensitivity to initial conditions, non-periodicity and non-convergence. However, attackers can exploit regions of regularity or low chaos within these systems to break encryption schemes, particularly if they possess knowledge of the plaintext used in encryption or can select it. Therefore, chaotic systems are frequently employed in conjunction with other technologies to enhance their level of security, such as DNA [4,5,6,7], genetic operations [8,9,10], compressive sensing [11,12,13], fractal sorting matrix [14], bit matrix [15], combinatorial design structures [16,17,18,19,20,21,22,23], and others.

Most concepts in combinational design theory are directly defined on finite sets, which can compensate for the limitations of applying chaos theory to cryptography. To date, many combinational design structures have been implemented in the field of cryptography, including Latin squares [16,17,18], orthogonal arrays [19], Latin cubes [20,21,22], Hadamard matrices [24], etc. Latin squares are commonly utilized in image encryption thanks to their distinctive properties of dispersion, uniformity, and abundance. These algorithms initially employ Latin squares to generate 1D maps for scrambling, then perform XOR operations directly on 2D images. However, they suffer from the drawbacks of low scrambling efficiency and susceptibility to specific attacks. The utilization of orthogonal Latin squares enables the direct construction of a 2D map, resulting in an enhanced scrambling efficiency [25]. Orthogonal arrays can be utilized to select appropriate rows for generating a pair of orthogonal Latin squares, albeit at the cost of requiring a relatively large storage space. Latin cubes are particularly useful in encryption, especially when dealing with more complex scenarios. Therefore, here we adopt another typical combinatorial design structure: the difference matrix. With its discreteness and approximate uniformity, it requires only a small storage space, making it an ideal candidate to serve as a pseudo-random sequence. Moreover, each difference matrix possesses a crucial property in that selecting any nonzero row generates a Latin square, while choosing any two non-zero rows produces a pair of orthogonal Latin squares. Therefore, a difference matrix can provide multiple pseudo-random sequences and a variety of scrambling methods, making it highly suitable for image encryption.

In recent years, the utilization of genetic algorithms (GA) in encryption technology has emerged as a prominent research frontier within the realm of image encryption. GAs are a type of optimization algorithm that operate on the principle of natural selection and evolution. Advantages such as simplicity, robustness, intrinsic parallelism, and self-adaptation have made them a popular choice in various fields, including cryptography, combinatorial optimization, and more. In 2012, Abdullah et al. put forward a hybrid model combining a GA and a chaotic function [8]. It employed the correlation coefficient as the fitness function. This approach has weaknesses against certain attacks, however, primarily due to its reliance on plaintext images and the utilization of identical chaotic sequences to encrypt different images. Thus far, various enhanced genetic algorithms have emerged for image encryption by leveraging the optimization capabilities of GAs to generate high-quality ciphertext images [26,27,28,29], optimizing chaotic sequences [30], and employing genetic operations directly for encryption [10,31,32]. Additionally, some approaches have integrated novel technologies along with GAs to improve the security and efficiency of the resulting cryptosystem [33,34,35]. In 2018, Pashakolaee et al. put forward a novel image encryption algorithm (IEA) named Hyper-chaotic Feeded GA (HFGA) [26]. This algorithm mainly uses a GA in combination with a hyperchaotic system to optimize cryptographic images and provide a secure decryption mechanism for authorized recipients. In the same year, Mozaffari proposed a parallel IEA on the basis of bit-plane decomposition [27]. In this approach, a GA was utilized to perform permutation and replacement steps through crossover and mutation operations. The resulting algorithm enables parallel processing of multi-bit-plane encryption, thereby enhancing the encryption speed and rendering it suitable for real-time applications. In 2019, Premkumar et al. put forward a secure composite 3D chaos-based image encryption algorithm utilizing a GA [31]. The algorithm was implemented by arithmetic crossover, multi-point crossover, and permutation of combinatorial mutation operators. In 2021, Zhang put forward a new IEA integrating genetic mutation and MLNCML [34]. The proposed encryption scheme employs DNA mutation and crossover operations by chaotic sequences. By integrating chaotic systems into the DNA operations, the scheme can effectively counter man-in-the-middle attacks, which are prevalent in traditional DNA addition and subtraction operations. In 2022, Qobbi proposed a novel encryption scheme combining a 2D logical mapping and genetic manipulations [10]. The algorithm’s security was guaranteed through the use of CBC with genetic manipulations. In 2022, Liang proposed an image cryptosystem combining the Fibonacci-Q matrix and GA [29]. The four-layer encryption framework employed to bolster the encryption security of this cryptosystem consists of diffusion, scrambling, diffusion, and optimization. In 2023, Bhowmik put forward an IEA employing a modified chaotic system integrated with DE and GA [30]. The GA was used to generate a unique key-string that was employed for the transposition process in the confusion stage. In summary, GAs represent a meta-heuristic optimization technique that can be applied to enhance the behavior of chaotic sequences or to improve the stochastic properties of ciphertext images during the encryption process. By introducing a GA into image encryption, it is possible to achieve a highly secure and efficient method for encrypting images.

To realize the advantageous properties of a difference matrix together with the optimization capabilities of a GA, we propose an efficient image cryptosystem utilizing both techniques. Difference matrices can serve as generators for Latin squares or pseudo-random sequences, while GA operations of crossover and mutation are employed to generate new individuals. This is followed by local optimization to achieve overall image optimization. The proposed cryptosystem first generates a difference matrix, which is then utilized for initial encryption. Subsequently, genetic operations are employed to cross and mutate the encrypted image blocks, followed by multiple rounds of optimal preservation to enhance the randomness of the final ciphertext image. The novel contributions of this research are summarized below.

We introduce a novel combinatorial structure for image encryption consisting of a difference matrix. It can be utilized to generate Latin squares or serve as a pseudo-random sequence while maintaining simplicity and requiring only a small amount of memory.The above structure is used to generate a novel row-scrambling method using a pair of orthogonal Latin squares.We define the formula for a single-point mutation. New individuals are then generated through crossover and mutation operations. Multiple rounds of local optimization using the optimal preservation strategy are used to further enhance the output.Our simulation results demonstrate the effectiveness of this cryptosystem in both encryption and decryption processes, exhibiting both robustness and practicality against conventional attacks.

The remaining sections of this paper are organized as follows: Section 2 presents a concise overview of the definitions for the difference matrix and the GA; Section 3 is dedicated to presenting the detailed processes for encryption and decryption; our simulation results and security analyses are presented in Section 4; finally, Section 5 serves to summarize this article.

## 2. Basic Definitions and Related Concepts

### 2.1. Difference Matrix

Let (G,+) be an Abel group of order *n*. An (n,k;1)-difference matrix is a k×n matrix $D=(d_{ij})$ with entries from *G* such that for any 1≤i<j≤k,  the multiset
$$\left\{d_{is}-d_{js}:1\leq s\leq n\right\}$$
(called the difference list) contains every element of *G* exactly once [36]. Removing any row from an (n,k;1)-difference matrix results in an (n,k−1;1)-difference matrix.

**Example** **1.**
*$D=\left(\begin{array}{ccc} 0 & 0 & 0\\ 0 & 1 & 2\\ 0 & 2 & 1 \end{array}\right)$ is a (3,3;1)-difference matrix over $Z_{3}$.*


A Latin square of order *n* (defined on an *n*-set *S*) is an n×n array in which each cell contains a single symbol; thus, each symbol occurs exactly once in each row and column. Two Latin squares of order *n*$A=(a_{ij})$ and $B=(b_{ij})$ are orthogonal if every ordered pair in S×S occurs exactly once in the juxtaposition array $C=(\left(a_{ij},b_{ij}\right))$.

**Theorem** **1****([36]).** *An (n,k;1)-difference matrix over an Abel group G gives rise to k−1 mutually orthogonal Latin squares of order n.*

**Theorem** **2****([37]).** *The multiplication table for the finite field $F_{q}$ is a (q,q;1)-difference matrix.*

Theorem 2 demonstrates that each nonzero row in a difference matrix can be used to generate a Latin square through the use of the addition operation. In addition, any two rows in a difference matrix can be used to generate a pair of orthogonal Latin squares.

**Example** **2.**
*Let $F_{4}$ be a field of order 4. Supposing that x is a primitive root of $F_{4}$ and that the primitive polynomial is $x^{2}+x+1$, let $g_{0}=0$, $g_{1}=1$, $g_{2}=x$, and $g_{3}=x+1$; then, $F_{4}=\left\{g_{0},g_{1},g_{2},g_{3}\right\}$.*


First, by utilizing Theorem 2 to generate the multiplication table of $F_{4}$, we can obtain a (4,4;1)-difference matrix $M_{t}$ with $M_{t}\left(i,j\right)=g_{i}\times g_{j}$. Subsequently, this matrix can be converted into digital format:$$M_{t}=\left(\begin{array}{cccc} 0 & 0 & 0 & 0\\ 0 & 1 & x & x+1\\ 0 & x & x+1 & 1\\ 0 & x+1 & 1 & x \end{array}\right)\to M_{t}=\left(\begin{array}{cccc} 0 & 0 & 0 & 0\\ 0 & 1 & 2 & 3\\ 0 & 2 & 3 & 1\\ 0 & 3 & 1 & 2 \end{array}\right).$$

By selecting two nonzero rows from the difference matrix $M_{t}$, namely, $row_{1}=2$ and $row_{2}=3$, we can form two Latin squares $Lm_{1}$ and $Lm_{2}$ using the addition operation in $F_{4}$:$$Lm_{1}=\left(\begin{array}{cccc} 0 & 1 & 2 & 3\\ 1 & 0 & 3 & 2\\ 2 & 3 & 0 & 1\\ 3 & 2 & 1 & 0 \end{array}\right),\;Lm_{2}=\left(\begin{array}{cccc} 0 & 2 & 3 & 1\\ 1 & 3 & 2 & 0\\ 2 & 0 & 1 & 3\\ 3 & 1 & 0 & 2 \end{array}\right).$$

We use $Lm_{1}$ and $Lm_{2}$ to construct the juxtaposition array $Lm_{12}$, where each ordered pair occurs only once. Thus, it can be inferred that $Lm_{1}$ and $Lm_{2}$ are orthogonal:$$Lm_{12}=\left(\begin{array}{cccc} 00 & 12 & 23 & 31\\ 11 & 03 & 32 & 20\\ 22 & 30 & 01 & 13\\ 33 & 21 & 10 & 02 \end{array}\right).$$

### 2.2. Logistic Map

The logistic map signal, which is one of the most well-known signals exhibiting chaotic behavior, can be described as follows (Equation 1):(1)$$x_{n+1}=\mu x_{n}\left(1-x_{n}\right),\;\;\;\;n=0,1,2,...$$
where $x_{n}$ is a real number between [0,1] and μ is a system parameter. When 3.573815<μ⩽4, the sequence $\left\{x_{n}|\;n=0,1,2,...\right\}$ has chaotic properties. Sensitivity to initial values and non-periodicity are the characteristics required by cryptography for keys and chaotic sequences.

### 2.3. Genetic Algorithm (GA)

Based on the principle of natural selection, John Holland first proposed GA as an optimization algorithm utilizing random search techniques in 1975. By selecting the most fit individuals and recombining their genetic information via crossover and mutation operations, a GA evolves a population of candidate solutions to generate new individuals. This process continues until satisfactory solutions are obtained or the algorithm reaches a predetermined stopping criterion. Thanks to their robustness, flexibility. and effectiveness in finding optimal solutions, GAs have been widely used in multiple optimization problems across different domains [38,39].

#### 2.3.1. Selection

A GA mainly evaluates individuals according to their fitness, with commonly employed methods including sorting selection, roulette wheel selection, etc. Occasionally, to prevent the loss of the best individuals, a strategy of optimal preservation is implemented whereby the fittest members of the population are directly carried over into the next generation without undergoing selection, crossover, or mutation.

#### 2.3.2. Crossover Operation

The crossover operation is a fundamental genetic operator in a GA that combines the genetic material of two or more parent individuals to generate new offspring. This process mimics the sexual reproduction found in nature, in which genes are inherited from parents. Various types of crossover operators exist within genetic algorithms, such as single-point, double-point, and multi-point crossovers, uniform crossover, arithmetical crossover, and many others. The crossover operation formula used in our proposed cryptosystem is presented below (Equation 2):(2)$$\begin{array}{c} \begin{array}{cc} & ns1=floor\left(ps1/2^{c}\right)\times 2^{c}+ps2\;\left(mod\;2^{c}\right),\\ & ns2=floor\left(ps2/2^{c}\right)\times 2^{c}+ps1\;\left(mod\;2^{c}\right), \end{array} \end{array}$$
where *c* denotes a crossover point, which is the index at which the genetic material of the two parent solutions is swapped; ps1 and ps2 represent two parent solutions, usually expressed as integers; ns1 and ns2 indicate two offspring solutions; and floor and mod represent the down round and module integer functions, respectively. We use (Equation 2) to formulate the crossover function (ns1,ns2) = crossover(ps1,ps2,c), which can be directly applied later. An illustrative example of the crossover operation is presented in Table 1.

#### 2.3.3. Mutation Operation

Mutation is another crucial genetic operator in GAs, introducing random changes to an individual’s genetic material to create new candidate solutions. This operator plays a vital role in maintaining population diversity, preventing premature convergence, and enhancing the algorithm’s local search capability. Various mutation operators are utilized in genetic algorithms, such as basic bit mutation, uniform mutation, Gaussian mutation, and others. In this paper, we define the rule of the mutation operation as follows (Equation 3):(3)$$gm=floor\left(pm/2^{c}\right)\times 2^{c}+2^{c}-1-pm\;\left(mod\;2^{c}\right)$$
where *c* is a mutation point, pm denotes a parent solution, and gm represents a newly generated solution. We use (Equation 3) to formulate the mutation function gm=mutation(pm,c) for direct application in subsequent steps. An illustrative example of the mutation operation is presented in Table 2.

## 3. The Proposed Cryptosystem

We now introduce several symbols that are utilized in the rest of this paper. The symbol *n* represents a prime power, *Q* denotes an n×n grayscale image, *C* signifies the corresponding ciphertext image, and $K_{1}$ refers to the initial encryption key. This cryptosystem is comprised of the following three parts. In Algorithm I, a difference matrix *M* defined on a finite field is generated using $K_{1}$. Algorithm II is utilized to perform the initial encryption, which consists of three layers: large-scale mutation, scrambling, and diffusion. This results in the formation of the initial encrypted image $C_{ini}$. In Algorithm III, both crossover and mutation operations are applied block-by-block to $C_{ini}$, and the final ciphertext image *C* is obtained over multiple iterations through the optimal preservation strategy. The encryption diagram of this cryptosystem is shown in Figure 1.

### 3.1. Generation of a Difference Matrix *M*

We generate a difference matrix *M* on a finite field. Initially, $K_{1}$ is utilized to produce an *n*-length chaotic sequence, which is employed to construct a finite field of order *n*. Subsequently, the difference matrix *M* is generated.

**Algorithm I:** The generation of *M*

**Input:** $K_{1}=\left(\mu_{0},key_{0},key_{1}\right)$.

**Output:** An (n,n;1)-difference matrix *M*.

**Step 1:** Generate a logistic sequence of length 1000+n using (Equation 1) with system parameter $\mu_{0}$ and initial value $x_{0}=key_{0}$. The sequence obtained by excluding the initial 1000 values is denoted as $X=\{x_{i}\;|\;i=0,1,2,...,n-1\}$. Use the function
(4)[fx,lx]=sort(X)
to sort *X* in ascending order and obtain a new sequence fx, and corresponding index subscript vector lx.

**Step 2:** Redefine the addition and multiplication operations on lx, then construct a finite field $F_{1}=\left\{g_{0},...,g_{n-1}\right\}$ [23]. generate an (n,n;1)-difference matrix *M* with $M\left(i,j\right)=g_{i}\times g_{j}$.

### 3.2. Generation of the Initial Ciphertext Image

This section describes the generation of the initial ciphertext image $C_{ini}$. First, a new encryption key is generated by adopting the information of the plaintext image. Then, a new *n*-length chaotic sequence is generated to construct the second finite field according to the method of generating a finite field from a sequence, as described in Algorithm I. Subsequently, a pair of orthogonal Latin squares is generated by selecting two nonzero rows from *M*. Finally, $C_{ini}$ is generated through three steps: large-scale mutation, scrambling, and diffusion.

**Algorithm II:** Generate the initial ciphertext image $C_{ini}$.

**Input:** An n×n grayscale image *Q*, $K_{1}=\left(\mu_{0},key_{0},key_{1}\right)$, *M* and a public parameter *a*.

**Output:** The initial ciphertext image $C_{ini}$.

**Step 1:** Read *Q* and calculate the sum of all the pixel values, denoted as sumQ. Letting
(5)$$s=floor\left(mod\left(sumQ,256\right)/256\times 10^{2}\right)/10^{2},$$
compute $key_{1\_new}=\left(key_{1}+s\right)/2$. Using $\mu_{0}$ as the system parameter and $key_{1\_new}$ as the initial value, generate the second *n*-length chaotic sequence, then form the second finite field $F_{2}$ using the method in Algorithm I.

**Step 2:** Obtain an (n,n−1;1)-difference matrix by removing the zero row of matrix *M*. Select rows mod(sumQ,n−1) and mod(sumQ+a,n−1) to generate two n×n orthogonal Latin squares $L_{m1}$ and $L_{m2}$ via $F_{2}$. By transforming these squares into row vectors, denoted as $L_{1}$ and $L_{2}$, respectively, two index indicators $ind_{1}$ and $ind_{2}$ can be constructed based on their uniformity using the following formula:(6)$$\begin{array}{c} \begin{array}{cc} & ind_{k}=mod\left(L_{k},8\right)+1,\;k=1,2. \end{array} \end{array}$$

**Step 3:** Large-scale mutation. Transform *Q* into a row vector $P_{1}$. Based on the parity of sumQ, distinct index indicators can be selected to sequentially mutate each element of $P_{1}$ and generate a new row vector $P_{2}$ using the mutation formula provided below.
(7)$$P_{2}\left(i\right)=mutation\left(P_{1}\left(i\right),ind_{2-mod(sumQ,2)}\left(i\right)\right),\;i=0,1,2,...,n^{2}-1.$$

**Step 4:** Image scrambling. Utilize the orthogonality of $L_{1}$ and $L_{2}$ to scramble $P_{2}$ and $L_{t}$ (The row vector formed by the transposition of *M*), resulting in the scrambled row vector $P_{3}$ and an approximately uniform vector $L_{T}$.
(8)$$\begin{array}{c} \begin{array}{cc} & P_{3}\left(i\right)=P_{2}\left(L_{1}\left(i\right)\times n+L_{2}\left(i\right)+1\right),\;i=0,1,2,...,n^{2}-1.\\ & L_{T}\left(i\right)=L_{t}\left(L_{2}\left(i\right)\times n+L_{1}\left(i\right)+1\right),\;i=0,1,2,...,n^{2}-1. \end{array} \end{array}$$

**Step 5:** Image diffusion. The initial ciphertext image $C_{ini}$ is formed using $L_{1}$, $L_{2}$ and $L_{T}$ as pseudo-random sequences for auxiliary diffusion.
(9)$$\begin{array}{c} \left\{\begin{array}{cc} & temp=mod(L_{1}\left(i\right)+L_{2}\left(i\right)+L_{T}\left(i\right),256),\\ & C_{ini}\left(i\right)=P_{3}\left(i\right)⊕temp⊕C_{ini}\left(i-1\right),\;i=0,1,2,...,n^{2}-1, \end{array}\right. \end{array}$$
where temp represents a temporary variable, $C_{ini}\left(-1\right)=0$ and ⊕ represents an XOR operation.

### 3.3. Optimization Process Using GA

In this section, we use a GA to optimize $C_{ini}$ in order to enhance the randomness of the final encrypted image. First, owing to the inherent randomness of GAs, multiple indexes are determined in order to restore the original plaintext image. Then, local crossover and mutation operations on $C_{ini}$ blocks are used to construct a large number of encrypted images, which significantly increases the level of randomness. Finally, after multiple rounds of genetic operations and using the optimal preservation strategy, the final encrypted image *C* and secondary key $K_{2}$ are generated.

**Algorithm III:** The optimization process using GA.

**Input:** $C_{ini}$, $ind_{1}$, $ind_{2}$, $L_{T}$, the number of crossover blocks $N_{c}$, the number of mutation blocks $N_{m}$, and the number of iterations Num.

**Output:** The final encrypted image *C* and the secondary key $K_{2}$.

**Step 1:** Process $L_{T}$ as follows to acquire the third index indicator $ind_{3}$:(10)$$ind_{3}=mod\left(L_{T},8\right)+1.$$

**Step 2:** Crossover operation. Divide $C_{ini}$ into $N_{c}$ parts from left to right and intersect each part according to $ind_{1}$ and $ind_{2}$ to form 2 ×$N_{c}$ new individuals. These individuals are then stored in the matrix $CM_{1}$. Using entropy as the fitness function, select the best individual, denoted as $C_{1}$, through the optimal preservation strategy. Simultaneously, record the corresponding index value $Ind_{cbest}$ in the first row component of $K_{2}$. The Algorithm 1 is presented below, where crossover represents the compiled crossover function (Equation 2).
**Algorithm 1** Local crossover operation1:**for all** $k=1:N_{c}$ **do**2:  **for all** $i=n^{2}/N_{c}*\left(k-1\right)+1:2:n^{2}/N_{c}*k-1$ **do**3:    $(CM_{1}\left(k,i\right),CM_{1}\left(k,i+1\right))=$ crossover$(C_{ini}\left(i\right),C_{ini}\left(i+1\right),ind_{1}\left(i\right));$4:    $(CM_{1}\left(N_{c}+k,i\right),CM_{1}\left(N_{c}+k,i+1\right))=$ crossover$(C_{ini}\left(i\right),C_{ini}\left(i+1\right),ind_{2}\left(i\right));$5:  **end for**6:**end for**7:$C_{1}=CM_{1}\left(Ind_{cbest},:\right);$8:**return** $C_{1}$

**Step 3:** Mutation operation. Divide $C_{1}$ into $N_{m}$ parts from left to right; each part undergoes mutation according to $ind_{2}$ and $ind_{3}$, resulting in the formation of 2 ×$N_{m}$ new individuals that are stored in the matrix $CM_{2}$. Use the optimal preservation strategy to select the best individual as $C_{2}$ and record the corresponding index value $Ind_{mbest}$ in the second row component of $K_{2}$. The Algorithm 2 is presented below, where mutation represents the compiled mutation function (Equation 3).
**Algorithm 2** Local mutation operation1:**for all** $k=1:N_{m}$ **do**2:  **for all** $i=n^{2}/N_{m}*\left(k-1\right)+1:1:n^{2}/N_{m}*k-1$ **do**3:    $CM_{2}\left(k,i\right)=$ mutation$(C_{2}\left(i\right),ind_{2}\left(i\right));$4:    $CM_{2}\left(N_{m}+k,i\right)=$ mutation$(C_{2}\left(i\right),ind_{3}\left(i\right));$5:  **end for**6:**end for**7:$C_{2}=CM_{2}\left(Ind_{mbest},:\right);$8:**return** $C_{2}$

**Step 4:** Let $C_{ini}=C_{2}$. Continue with Steps 2 and 3 until the number of iterations Num or $Ind_{cbest}$ and $Ind_{mbest}$ match that of the previous column in $K_{2}$, then terminate the calculation and record final iteration number as $Ind_{best}$.

**Step 5:** Transpose $C_{2}$, the vector obtained after the iteration, into an n×n matrix in order to obtain *C* and output $K_{2}$.

### 3.4. Image Decryption Process

The proposed cryptosystem employs asymmetric processes for both encryption and decryption. To decrypt *C*, the receiver requires $K_{1}$ and $K_{2}$ along with the characteristic value sumQ and the public parameter *a*. The decryption diagram of this cryptosystem is illustrated in Figure 2.

The steps are described in detail below.

**Step 1:** First, $K_{1}$ is utilized to generate *M* following Algorithm I. Then, three row vectors $L_{1}$, $L_{2}$, and $L_{T}$ are generated using sumQ and *a*.

**Step 2:** *C* is transformed into a row vector $R_{1}$ and the pre-optimized image $R_{2}$ is decrypted using $K_{2}$. This step is asymmetric with regard to the optimization process, and can be decrypted directly by using $K_{2}$. Initially, mutation is performed according to the second index block of the last column of $K_{2}$, then crossover is conducted according to the first index block, iterating $Ind_{best}$ times to obtain $R_{2}$.

**Step 3:** The original image *R* is recovered from $R_{2}$. First, reverse diffusion is performed on $R_{2}$ to obtain $R_{3}$, then reverse scrambling to obtain $R_{4}$, followed by reverse large-scale mutation to acquire $R_{5}$. Finally, the result is transposed into an n×n matrix in order to obtain the decrypted image *R*. This step is a complete reversal of the initial encryption process, and is not repeated.

## 4. Simulation Results and Safety Analyses

To validate the security and efficacy of the cryptosystem proposed in this paper, a set of 256×256 grayscale images from the USC-SIPI2 image set were selected as test samples. The initial key $K_{1}$ was set to $\mu_{0}=3.99999$, $key_{0}=0.123456$ and $key_{1}=0.234567$, whilehe public parameters were set to a=31, $N_{c}=16$, $N_{m}=32$, and Num=20.

In the process of utilizing GAs for image encryption, fitness functions such as entropy and correlation coefficients are commonly employed for assessing the quality of ciphertext images. However, due to the stochastic nature of correlation coefficients, in this paper we have instead adopted the information entropy of the encrypted image as the fitness function.

We evaluated our proposed cryptosystem through simulation experiments from various aspects, including key space and sensitivity analyses, statistical analyses, resistance to differential attacks, and more. To demonstrate its superiority over recent algorithms, a comparison is provided in terms of security and efficiency.

### 4.1. Simulation Results

Our cryptosystem is highly effective in converting a grayscale image into a meaningless encrypted image, thereby ensuring the confidentiality and privacy of the original. Additionally, the decrypted image remains unchanged, with no loss or distortion, guaranteeing its integrity and authenticity. Figure 3a shows the several plaintext images, Figure 3c depicts the corresponding ciphertext images, and Figure 3e illustrates the lossless decrypted images. In this cryptosystem, the initial encryption process ensures that any image meets the requirements for encryption, while the optimization stage further enhances the overall randomness. The second keys and the iteration numbers required for optimization are displayed in Table 3. For a majority of the images, the optimization process can be completed within 20 generations, resulting in relatively higher-quality ciphertext images. 

### 4.2. Key Analyses

#### 4.2.1. Key Space Analysis

The key space of a cryptosystem is a metric describing its resistance against brute-force attacks. A larger key space results in increased resistance to such attacks, making it more difficult for attackers to determine the correct key. The initial key $K_{1}=\left(\mu_{0},key_{0},key_{1}\right)$ has a precision of $10^{-15}$ for each real number. Therefore, the key space can reach at least $10^{45}\approx 2^{149}$, which is significantly larger than $2^{128}$ [40,41]. In addition, each image has a unique $K_{2}$ value during decryption, and there is a public parameter that can be selected. Therefore, the key space of the proposed cryptosystem is of sufficient size to defend against brute-force attacks.

#### 4.2.2. Key Sensitivity Analyses

Key sensitivity is a crucial property of an image cryptosystem, ensuring that even the slightest alteration in the encryption or decryption key will result in two entirely distinct datasets. In regard to the proposed cryptosystem, the key sensitivity can be analyzed from two perspectives: during the encryption stage and during the decryption stage.

To analyze the key sensitivities of the proposed cryptosystem, we selected Lena as the test image. The resulting ciphertext image when encrypting Lena with the initial key $K_{1}=\left(3.99999,0.123456,0.234567\right)$ is denoted as cipher1. A minute value of $10^{-15}$ was subsequently added to each element of $K_{1}$, resulting in cipher2 as the ciphertext image obtained by encrypting Lena with the altered key. A comparison of the results of two encrypted images under three sets of parameters is shown in Figure 4, revealing that while both images contain noise, they differ significantly from each other. Table 4 calculates the percentages of different pixels in the two contrasting images after encryption, with the results surpassing 99.6%. This fully demonstrates the exceptional sensitivity of the proposed cryptosystem during encryption.

For the sensitivity analysis performed during decryption, we again used Lena as the test image. Utilizing $K_{1}=\left(3.99999,0.123456,0.234567\right)$ and $K_{2}$ to decrypt *C*, the original image can be obtained; however, even a slight alteration of $10^{-15}$ in each value of $K_{1}$ results in a meaningless and completely different decryption compared to the original image. The contrasting results are presented in Figure 5. Further, Table 4 lists the percentages of different pixels between *Q* and the decryption using the modified key, reaching more than 99.52%. This fully demonstrates that the proposed cryptosystem is highly sensitive to $K_{1}$. Additionally, there is an equally strong sensitivity to $K_{2}$, which need not be reiterated here.

### 4.3. Histogram Analyses

In the context of image encryption algorithms, the randomness and uniformity of the encrypted image are important for protecting against statistical attacks. The more random and uniform the ciphertext image is, the more challenging it becomes for attackers to identify correlations or patterns that can be exploited to break the encryption [40]. Typically, the variance *S* is employed as to describe the uniformity of the pixel distribution. Its calculation formula is
(11)$$S=\frac{1}{256}\sum_{i=0}^{255}{\left(hist_{i}-aver\right)}^{2},$$
where $hist_{i}$(i=0,1,...,255) represents the occurrence of the *i*th grayscale value in the image and aver denotes the mean value of $hist_{i}$. A smaller value of *S* implies a more homogeneous distribution of grayscale pixels in the encrypted image, posing a challenge for attackers to exploit any statistical patterns and break the encryption.

Figure 3 illustrates the changes in the histograms of multiple images pre- and post-encryption. The histograms of the ciphertext images display a more homogeneous distribution in comparison to that of the original image, indicating an approximately equal frequency of occurrence for each grayscale value. Table 5 presents the variance values of multiple images pre- and post-encryption. The data indicate that after encryption, the variance values decrease significantly and are all smaller than the standard value $\chi_{0.05}^{2}=293.25$, which meets the encryption requirement. Notably, the variance of the encrypted Lena image is as low as 101.133, indicating the effectiveness of our proposed cryptosystem in resisting histogram analyses.

### 4.4. Correlation Coefficient Analyses

In a plaintext image, adjacent pixels exhibit a strong correlation, which can be exploited by attackers to break the encryption using statistical patterns or correlations. Therefore, an effective encryption algorithm should eliminate the correlation of adjacent pixels in order to increase resistance against statistical analyses. Typically, correlation coefficients are commonly computed to assess the strength of correlations using the formula
(12)$$r_{uv}=\frac{E\left[u-E(u)][v-E(v)\right]}{\sqrt{D(u)}\sqrt{D(v)}},$$
(13)$$E\left(u\right)=\frac{1}{N}\sum_{i=1}^{N}u_{i},D\left(u\right)=\frac{1}{N}\sum_{i=1}^{N}{\left(u_{i}-E\left(u\right)\right)}^{2},$$
where *u* and *v* represent the gray values of two adjacent pixels, E(u) denotes the expected value of *u*, D(u) represents its variance, and *N* denotes the quantity of samples randomly selected from the image.

In this study, we selected 4000 pairs of adjacent pixels in a stochastic manner from the vertical, horizontal, and diagonal directions in the image. The coefficients in the three directions *x*, *y*, and *z* were then calculated according to Equations (Equation 12) and (Equation 13) and the average coefficients were obtained using Equation (Equation 14).
(14)$$aver_{coef}=\frac{\left|x|+|y|+|z\right|}{3}$$

The resulting coefficients are presented in Table 5. In the table, the correlation coefficient of each plaintext image is highly proximate to 1 in three distinct directions, indicating a strong correlation between adjacent pixels. Conversely, the correlation coefficient of every ciphertext image approaches 0, signifying a weakened correlation. To demonstrate the optimization effect of our cryptosystem, we compared the correlation coefficients of the encrypted Lena image with those obtained from several recently proposed algorithms. The contrasting results of the coefficients are displayed in Table 6. As can be seen, the average correlation coefficient when using our cryptosystem is larger than the results for those in [29,42], but smaller than those in the other seven papers. This indicates that our approach can effectively mitigate the pixel correlation.

To obtain a more intuitive visual comparison, Figure 6 displays the pre- and post-encryption distribution of neighboring pixels in three directions for three different images when using our proposed cryptosystem. The midpoint of the original grayscale image is effectively scattered along the diagonal line, whereas that of the corresponding ciphertext image is more uniformly distributed within the entire scope. This demonstrates that the proposed cryptosystem is capable of achieving superior confounding effects and successfully passing the correlation test.

### 4.5. Entropy Analyses

Information entropy is another crucial measure for testing the uncertainty or randomness of ciphertext images, and can also be used to assess the efficacy of encryption algorithms. Its symbolic representation is H, and the calculation formula can be expressed as follows:(15)$$H\left(m\right)=-\sum_{i=0}^{l-1}p\left(m_{i}\right)log_{2}p\left(m_{i}\right)$$
where *l* is the number of gray values and $m_{i}$ and $p(m_{i})$ are the gray values and their occurrence probabilities, respectively. In cryptography, a secure cryptosystem should produce ciphertext images with uniform gray values in order to increase randomness and security. The ideal value of information entropy for an image with 256 gray levels is 8 [46].

Following Equation (Equation 15), we calculated the information entropy values of multiple encrypted images. Table 5 displays the results, while Table 6 lists the contrasting results obtained for Lena with different algorithms. As can be seen, all entropy values are very close to 8; the information entropy for the encrypted Lena image reaches 7.99888, which is a larger value than most of the other algorithms. Therefore, the proposed image cryptosystem is capable of generating ciphertext images with a more random and uniform distribution of pixel values, resulting in increased security and confidentiality.

### 4.6. Differential Attack Analyses

In order to be resistant to differential attacks, a secure cryptosystem should guarantee that even minor alterations to the plaintext will lead to large changes in the encrypted image. In general, two criteria, NPCR and UACI, are used to evaluate resistance to differential attacks. The respective formulas are provided below.
(16)$$NPCR=\frac{\sum_{i=0}^{M-1}\sum_{j=0}^{N-1}D\left(i,j\right)}{M\times N}\times 100\%,$$
where $D_{i,j}=\left\{\begin{array}{cc} & 0,\;\;C_{1}\left(i,j\right)=C_{2}\left(i,j\right)\\ & 1,\;\;C_{1}\left(i,j\right)\neq C_{2}\left(i,j\right) \end{array}\right.$
(17)$$UACI=\frac{\sum_{i=0}^{M-1}\sum_{j=0}^{N-1}\left|C_{1}\left(i,j\right)-C_{2}\left(i,j\right)\right|}{255\times M\times N}\times 100\%.$$

In the above formulas, $C_{1}$ and $C_{2}$ are two ciphertext images that correspond to plaintext images with only one different pixel, while *M* and *N* represent the width and height of the encrypted image, respectively.

For each plaintext image, 100 pixels at different positions are randomly selected and increased by 1. Table 7 displays the maximum, lowest, and average values of NPCR and UACI. The significance level was set to 0.05, and the ideal expected values of NPCR and UACI for grayscale images with a size of 256×256 were obtained from [47]. The average NPCR values in Table 7 exceed the ideal values. Moreover, the average values of UACI are found to fall within the interval [$U_{0.05}^{*-}$, $U_{0.05}^{*+}$]. This testifies that the proposed image cryptosystem exhibits high resistance to differential attacks with respect to pixel modifications.

### 4.7. Cutting and Noise Attack Analyses

Signal loss or noise pollution can compromise the security and confidentiality of transmitted ciphertext images. To test a ciphertext image’s resilience against data loss and noise attacks, experiments can simulate such conditions to assess the encryption algorithm’s ability to maintain data integrity [48].

Cutting parts of the ciphertext image can cause information loss and hinder the successful recovery of the original plaintext message. Taking the Plane image as an example, the corresponding decryptions are shown in Figure 7 when cutting 1/16, 1/8, 1/4, and 1/2 at the upper left corner of the encrypted image. As can be seen, even when up to 50% of the encrypted image is cut, the proposed cryptosystem is still able to recover a visually recognizable image of the original message using the correct key. In addition, it demonstrates strong robustness against various types of noise attacks, including salt and pepper noise (SPN) with densities of 0.05 and 0.1 and Gaussian noise (GN) with a mean value of 0 and variances of 0.01 and 0.1. Despite the introduction of such noise types into the ciphertext image, the deciphering process utilizing the correct encryption key remains discernible, as demonstrated in Figure 8. Overall, these results demonstrate the good robustness of our cryptosystem against both cutting attacks and different types of noise attacks.

Additionally, the PSNR index is usually employed for assessing the quality of decrypted images after attacks. The formula is
(18)$$\mathrm{PSNR}=10\times log_{10}\frac{M\times N\times 255^{2}}{\sum_{i=0}^{M-1}\sum_{j=0}^{N-1}{\left(P_{1}\left(i,j\right)-P\left(i,j\right)\right)}^{2}},$$
where *P* and $P_{1}$ denote the original decrypted image of size M×N and the decrypted image after attacks, respectively. A higher PSNR value indicates better image quality.

Table 8 displays the PSNR values of various images that have undergone cutting and noise attacks using the proposed encryption cryptosystem. The data in the table suggest that the PSNR values under these cutting attacks are greater than 7.7 dB, while those for the noise attack are greater than 7.2 dB. These findings suggest that the decrypted images exhibit relatively low levels of distortion compared to their original counterparts even after being attacked, and continue to maintain a high level of visual quality. Therefore, these results indicate that our cryptosystem is robust against both cutting and noise attacks.

### 4.8. Computational Complexity and Encryption Speed Analyses

Computational complexity (CC) and encryption speed are crucial factors to consider when evaluating the validity of an image cryptosystem. First, Algorithm I generates a chaotic sequence with length *n*, with computational complexity O(n), followed by constructing a difference matrix with computational complexity O$\left(n^{2}\right)$. In Algorithm II, there are three layers of encryption structure: large-scale mutation, scrambling, and diffusion; thus, the CC is O$\left(3n^{2}\right)$. Algorithm III is an optimization process, including crossover and mutation operations, meaning that the CC is also O$\left(4n^{2}\right)$. Therefore, the CC of our cryptosystem is O$\left(n^{2}\right)$.

The experimental environment was MATLAB R2019b, Microsoft Windows 10, Intel i5-1135G7, a 2.40 GHz processor, and 16 GB RAM. Based on 100 calculations, Table 9 presents the average time taken to perform encryption and decryption of six representative images. The results indicate that each image takes approximately 5.86 s to encrypt and about 0.43 s to decrypt, suggesting the effectiveness of our image cryptosystem for instant encryption. The proposed cryptosystem employs an asymmetric process for encryption and decryption that involves multi-segment and multi-round crossover and mutation operations during encryption, which can be computationally expensive and result in longer optimization and encryption times compared to other algorithms. However, the decryption time is significantly shorter when using the secondary secret key to decrypt the ciphertext image directly. The comparison results reported in Table 9 demonstrate that our GA-based encryption cryptosystem is faster in both encryption and decryption compared to other contrasting algorithms, with the exception of the algorithm presented in [29], reflecting the efficient optimization process of the proposed cryptosystem.

### 4.9. Analysis of Resistance to Chosen-Plaintext Attacks

Chosen-plaintext attacks are a potent method of attacking encryption schemes. In such an attack, the attacker has access to the encryption scheme, and can select plaintext images and encrypt them to obtain the key’s information. If a cryptosystem is able to withstand chosen-plaintext attacks, this usually indicates that the encryption approach is secure against three other classical attacks as well: ciphertext-only attacks, known-plaintext attacks and chosen-ciphertext attacks [21,53].

In the proposed cryptosystem, the same difference matrix is used for all images in image encryption. This matrix only needs to be generated once, and can be reused to encrypt multiple images. However, employing the information of plaintext images in the key ensures that the encrypted data remain unique and secure for every individual image even if the difference matrix is identical. Specifically, the entire cryptosystem is extremely sensitive to both $K_{1}=\left(\mu_{0},key_{0},key_{1}\right)$ and to the plaintext image. If even a single pixel is altered, the resulting $L_{1}$, $L_{2}$, and $L_{T}$ will be completely distinct. Additionally, our encryption cryptosystem utilizes two Latin squares and leverages the approximate uniformity of the difference matrix during diffusion. With just one round of encryption processing required to achieve a secure result, our cryptosystem is resistant to chosen-plaintext attacks, as well as other forms of intrusion.

Additionally, all-black or all-white images are frequently utilized by hackers to launch attacks on encryption algorithms. These special images were also subjected to experiments. Table 5 shows the resulting correlation coefficients of approximately 0, variance values of approximately 100, and entropies of approximately 8 for two encrypted ciphertext images. These results fully demonstrate that the proposed cryptosystem can resist statistical attacks. Table 7 shows that both of the encrypted images pass the differential attack tests successfully. Table 8 displays that all PSNR values related to both encrypted images exceed 7.2 dB, demonstrating our cryptosystem’s robustness. Moreover, our cryptosystem exhibits resistance against chosen-plaintext attacks as well as other classical attack methods.

## 5. Conclusions

In this paper, an efficient image cryptosystem utilizing a difference matrix and a genetic algorithm is put forward. The discreteness and approximate uniformity of the difference matrix satisfy the characteristics of a pseudo-random sequence, enabling a variety of scrambling methods while occupying a very small storage space. Additionally, the GA’s crossover and mutation operations can generate numerous new individuals, and the optimal preservation strategy can then select the best one. The proposed cryptosystem first generates a difference matrix using the initial secret key; then, the difference matrix is used to perform image encryption. The resulting initial ciphertext image already has relatively good randomness. A large number of ciphertext images are then obtained through local genetic operations in order to generate better-quality ciphertext images through multiple iterations using the optimal preservation strategy, which strengthens the randomness of the final ciphertext image. We used this cryptosystem to effectively encrypt test images, and it successfully passed all tests. Furthermore, it offers a number of advantages over other existing algorithms.

Finally, the use of a combined structure consisting of a difference matrix and a genetic algorithm to generate more secure ciphertext images showcases interdisciplinary advantages that can serve as a reference for future research.

## Figures and Tables

**Figure 1 entropy-26-00351-f001:**
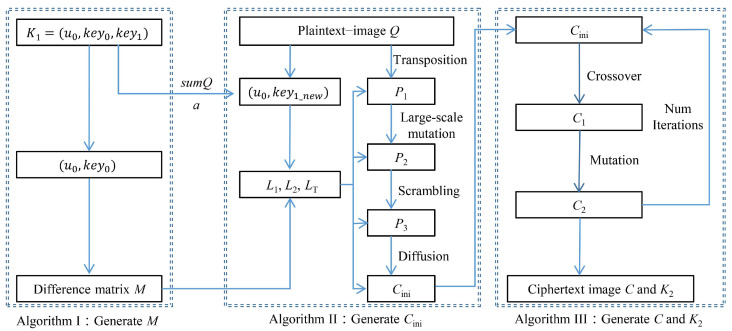
Encryption diagram of the proposed cryptosystem.

**Figure 2 entropy-26-00351-f002:**
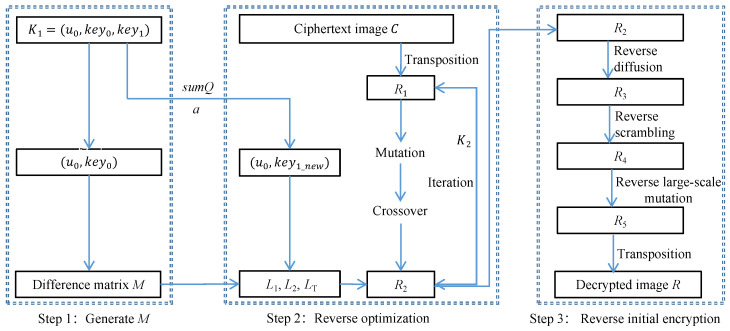
Decryption diagram of the proposed cryptosystem.

**Figure 3 entropy-26-00351-f003:**
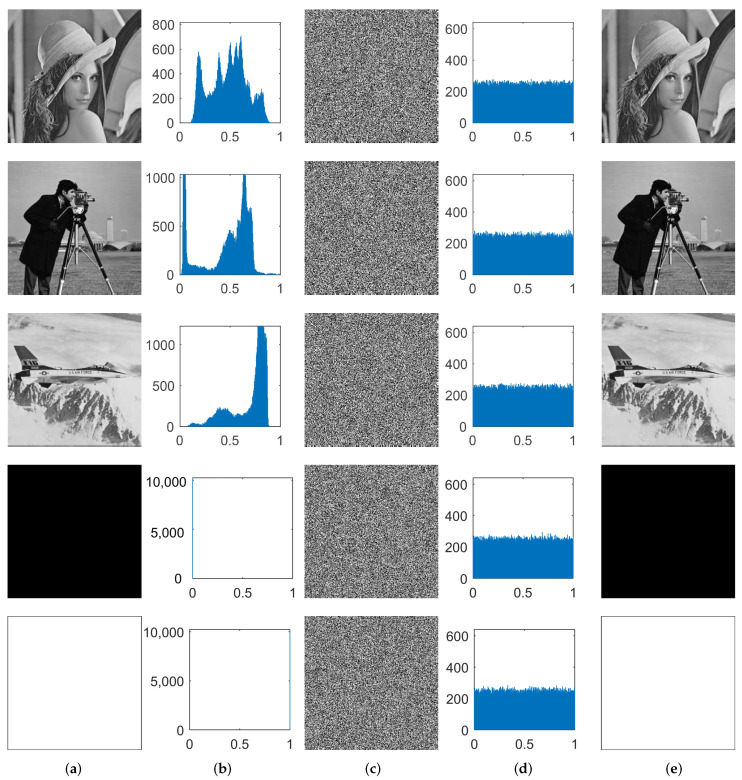
Histograms of different plaintext images and ciphertext images. The plaintext images from top to bottom are as follows: Lena, Cameraman, Plane, Allblack, Allwhite. (**a**) Plaintext images; (**b**) the corresponding histograms of (a); (**c**) ciphertext images; (**d**) the corresponding histograms of (**c**); (**e**) the decryption of (**c**).

**Figure 4 entropy-26-00351-f004:**
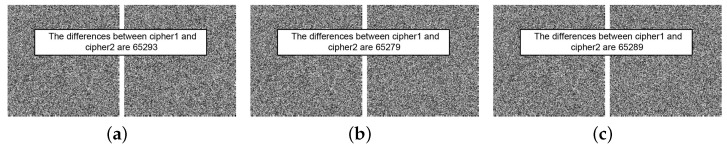
Key sensitivity test results in the encryption stage: (**a**) $\mu_{0}+10^{-15}$, (**b**) $key_{0}+10^{-15}$, and (**c**) $key_{1}+10^{-15}$.

**Figure 5 entropy-26-00351-f005:**
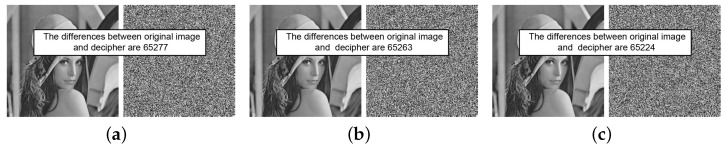
Key sensitivity test results in the decryption stage: (**a**) $\mu_{0}+10^{-15}$, (**b**) $key_{0}+10^{-15}$, and (**c**) $key_{1}+10^{-15}$.

**Figure 6 entropy-26-00351-f006:**
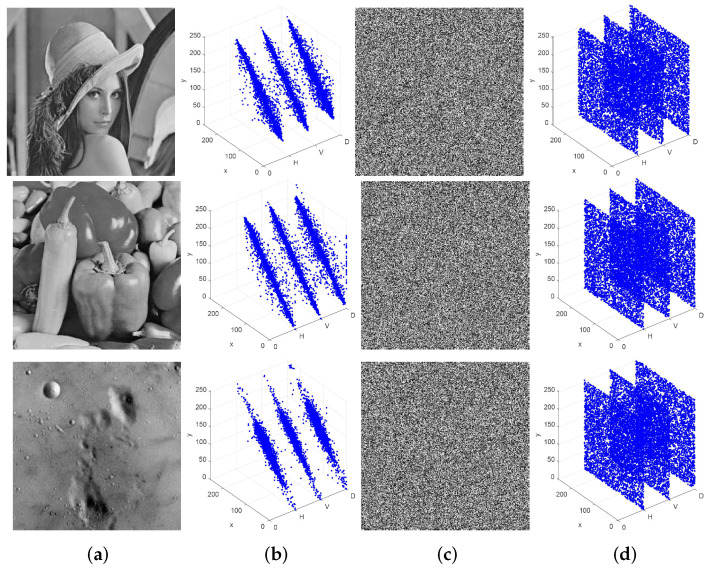
Distribution of adjacent pixels of Lena, Pepper, and 5.1.09: (**a**) Plaintext images; (**b**) Distribution of adjacent pixels of (**a**); (**c**) ciphertext images; (**d**) Distribution of adjacent pixels of (**c**).

**Figure 7 entropy-26-00351-f007:**
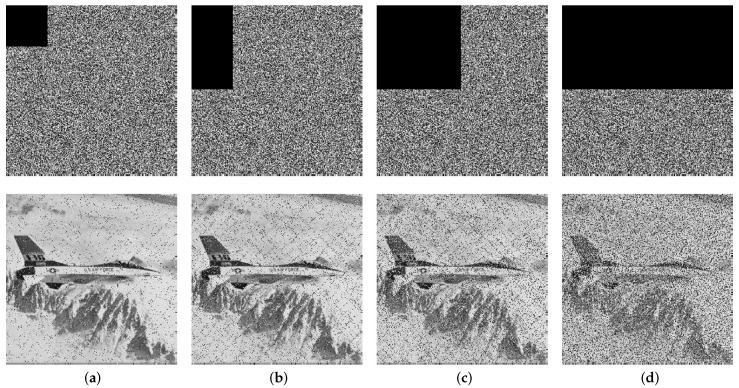
Ciphertext Plane images under different cutting levels and the corresponding decryptions: (**a**) 1/16; (**b**) 1/8; (**c**) 1/4; (**d**) 1/2.

**Figure 8 entropy-26-00351-f008:**
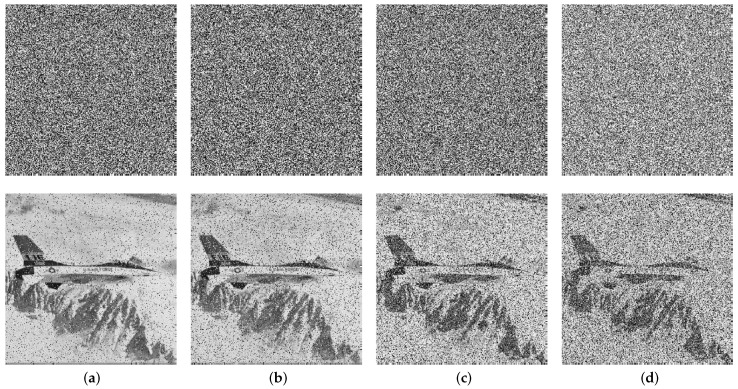
Ciphertext Plane images under noise attacks and the corresponding decryptions: (**a**) SPN (0.05); (**b**) SPN (0.1); (**c**) GN (0.01); (**d**) GN (0.1).

**Table 1 entropy-26-00351-t001:** An example of a crossover operation with c=3.

Pixel Value	Decimal Representation	Binary Representation
ps1	178	10110 | 010
ps2	46	00101 | 110
ns1	182	10110 | 110
ns2	42	00101 | 010

**Table 2 entropy-26-00351-t002:** An example mutation operation with c=6.

Pixel Value	Decimal Representation	Binary Representation
pm	178	10 | 110010
gm	141	10 | 001101

**Table 3 entropy-26-00351-t003:** The final results of the iteration numbers and $K_{2}$ required for optimization.

Image	Iteration Number	$K_{2}$
Lena	19	17 25 12 5 30 30 13 10 26 13 13 12 2 4 4 7 1 1 22
		15 26 64 12 61 28 22 19 47 45 20 39 27 31 44 11 52 43 52
Cameraman	14	16 26 24 17 29 24 14 14 1 31 12 31 19 31
		3 38 51 10 52 22 12 28 30 24 26 45 38 41
Plane	13	6 16 13 20 18 22 11 32 28 17 18 2 1
		44 15 22 32 19 5 36 11 24 15 36 15 15
Pepper	18	15 11 17 19 18 30 24 10 13 18 24 14 14 14 30 29 13 13
		46 20 22 17 32 18 27 3 58 15 43 44 30 55 57 12 55 6
5.1.09	15	17 18 3 22 19 14 20 20 7 23 29 29 4 23 17
		48 28 49 59 6 52 40 45 4 5 46 60 58 46 46
6.1.01	13	31 9 24 17 32 18 2 11 5 5 12 7 22
		12 7 52 58 30 16 43 31 11 42 24 13 24
6.2.01	11	22 4 32 24 27 30 26 18 32 8 12
		15 37 22 27 50 24 28 62 27 32 42
Allwhite	20	13 8 26 6 31 18 23 3 7 28 11 11 11 29 28 20 31 19 31 12
		8 33 50 34 45 39 35 19 56 63 47 29 61 25 46 63 37 57 24 55
Allblack	20	20 24 22 26 13 16 7 32 7 31 27 5 27 13 27 27 27 21 7 27
		37 31 25 29 64 56 6 52 61 61 57 62 9 33 1 15 46 47 56 54

**Table 4 entropy-26-00351-t004:** Number and percentage changes during encryption and decryption.

Image	Encryption			Decryption		
	Figure 4a	Figure 4b	Figure 4c	Figure 5a	Figure 5b	Figure 5c
The number of different pixels	65,293	65,279	65,289	65,277	65,263	65,224
Percentage (%)	99.6292	99.6078	99.6231	99.6048	99.5834	99.5239

**Table 5 entropy-26-00351-t005:** Complete results of several images pre- and post-encryption.

Image	Testing Direction			Average Value	Variance	Entropy
	H	V	D			
Lena	0.9410	0.9654	0.9221	0.9428	$4.14\times 10^{4}$	7.42489
Ciphertext image Lena	0.0010	0.0029	-0.002	0.0020	101.133	7.99888
Cameraman	0.9553	0.9735	0.9406	0.9565	$1.06\times 10^{5}$	7.03056
Ciphertext image Cameraman	0.0025	−0.0005	−0.0036	0.0022	117.188	7.99871
Plane	0.9426	0.9249	0.8833	0.9169	$1.75\times 10^{5}$	6.72033
Ciphertext image Plane	0.0005	0.0043	0.0013	0.0020	117.82	7.9987
Pepper	0.9547	0.9542	0.9041	0.9376	$3.67\times 10^{4}$	7.53524
Ciphertext image Pepper	−0.0024	0.0069	0.0039	0.0044	119.22	7.99869
5.1.09	0.9040	0.9360	0.9000	0.9134	$1.36\times 10^{5}$	6.70931
Ciphertext image 5.1.09	0.0006	−0.0032	−0.0034	0.0024	124.813	7.99863
6.1.01	0.9862	0.9917	0.9738	0.9839	$1.22\times 10^{5}$	7.20445
Ciphertext image 6.1.01	−0.0014	0.0025	0.0006	0.0015	112.344	7.99876
6.2.01	0.9424	0.9105	0.8818	0.9116	$6.33\times 10^{4}$	7.16791
Ciphertext image 6.2.01	0.0013	0.0026	−0.0027	0.0022	133.18	7.99853
Allwhite	NaN	NaN	NaN	NaN	$1.67\times 10^{7}$	0
Ciphertext image Allwhite	0.0029	0.0025	−0.0001	0.0018	108.656	7.99881
Allblack	NaN	NaN	NaN	NaN	$1.67\times 10^{7}$	0
Ciphertext image Allblack	−0.0019	0.0018	−0.0008	0.0015	104.469	7.99886

**Table 6 entropy-26-00351-t006:** Contrasting results for Lena.

Image	Testing Direction			Average Value	Entropy	NPCR (%)	UACI (%)
	H	V	D				
The proposed cryptosystem	0.0010	0.0029	−0.0020	0.0020	7.99888	99.6081	33.581
Lena in [10]	0.0019	0.0014	0.0052	0.0028	7.9992	99.614	33.364
Lena in [12]	0.0076	0.0093	−0.016	0.011	7.9995	99.65	33.64
Lena in [27]	−0.00058	0.0048	−0.0243	0.0099	7.9968	99.58	33.08
Lena in [29]	−0.0063	0.0022	−0.0006	0.003	7.9985	99.61	33.48
Lena in [34]	−0.0239	−0.0033	0.0046	0.0106	7.9966	99.64	33.48
Lena in [42]	−0.0026	−0.0012	−0.0011	0.0016	7.9976	99.59	33.43
Lena in [43]	0.0056	0.0037	0.0032	0.0042	7.9976	99.62	33.4169
Lena in [44]	0.0305	−0.0043	0.0042	0.013	7.9976	99.61	33.51
Lena in [45]	0.0015	−0.0026	0.0042	0.0028	7.9976	99.6002	33.4592

**Table 7 entropy-26-00351-t007:** NPCR and UACI results for several test images.

Image	NPCR (%)			Results		UACI (%)			Results		
	max	min	mean	$N_{0.05}^{*}$	**Results**	**max**	**min**	**mean**	$U_{0.05}^{*-}$	$U_{0.05}^{*+}$	**Results**
Lena	99.6643	99.5483	99.6081	99.5693	Pass	33.6895	33.448	33.581	33.2824	33.6447	Pass
Cameraman	99.6765	99.5514	99.6184	99.5693	Pass	33.6778	33.4022	33.5571	33.2824	33.6447	Pass
Pepper	99.6674	99.5499	99.6043	99.5693	Pass	33.5379	33.3183	33.4285	33.2824	33.6447	Pass
Plane	99.6521	99.5438	99.6003	99.5693	Pass	33.5982	33.3891	33.4953	33.2824	33.6447	Pass
5.1.09	99.6643	99.5377	99.6109	99.5693	Pass	33.4543	33.2354	33.3726	33.2824	33.6447	Pass
6.1.01	99.675	99.5743	99.6183	99.5693	Pass	33.7306	33.4918	33.6222	33.2824	33.6447	Pass
6.2.01	99.6735	99.5636	99.6175	99.5693	Pass	33.5592	33.31	33.4586	33.2824	33.6447	Pass
Allwhite	99.7589	99.5407	99.6434	99.5693	Pass	33.7763	33.2249	33.4591	33.2824	33.6447	Pass
Allblack	99.6689	99.5682	99.6287	99.5693	Pass	33.4928	33.334	33.4198	33.2824	33.6447	Pass

**Table 8 entropy-26-00351-t008:** All PSNR values for robustness analyses of multiple images.

Image	PSNR (dB)				PSNR (dB)			
	Cut 1/16	Cut 1/8	Cut 1/4	Cut 1/2	SPN(0.05)	SPN(0.1)	GN(0.01)	GN(0.1)
Lena	21.3081	18.2894	15.2783	12.2309	19.2083	16.5939	13.0183	11.9036
Cameraman	20.4213	17.3969	14.3934	11.4299	18.4421	15.5148	12.288	11.1186
Plane	19.9419	16.8564	13.9526	11.0007	18.1729	15.4114	11.9437	10.7354
Pepper	20.8825	17.9003	14.8761	11.955	18.9812	16.1368	12.9661	11.7686
5.1.09	22.2761	19.2009	16.1995	13.216	20.1598	17.2207	13.503	12.4539
6.1.01	20.1103	17.1455	14.1046	11.1382	18.2645	15.3268	11.736	10.7258
6.2.01	20.8657	17.8207	14.8093	11.802	18.8325	15.8396	12.6107	11.4927
Allwhite	16.7344	13.7197	10.739	7.7516	14.9826	12.1055	8.1539	7.226
Allblack	16.8237	13.7882	10.7569	7.7957	15.0808	12.295	8.1306	7.2233

**Table 9 entropy-26-00351-t009:** Time efficiency results for several images.

Image	Encryption Time(s)	Decryption Time(s)
Lena	7.2093	0.4303
Cameraman	5.3426	0.4105
Plane	5.1036	0.4116
Pepper	6.8045	0.528
5.1.09	5.7075	0.4088
6.1.01	4.9968	0.4122
**Average**	**5.86**	**0.43**
Encrypted Lena in [49]	10.9	–
Encrypted Lena in [50]	14.1	–
Encrypted Lena in [51]	6.9	–
Encrypted Lena in [52]	7.07	–
Encrypted Lena in [29]	0.88	–

## Data Availability

The data used to support the findings of this study are included within the article.

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
