# Peer review of "An Efficient Image Cryptosystem Utilizing Difference Matrix and Genetic Algorithm"

_entropy, 2024, doi:10.3390/e26050351_

Round 1
Reviewer 1 Report
Comments and Suggestions for Authors
Authors proposed an image encryption method based on difference matix, logistic map and genetic algorithm. Compared with other methods, good performance can be achieved. There are some minor points to be answered as follows:
1. In Fig. 4 and 5, the number of different pixels are given in figures. It seems not necessary, because the indicators such as NPCR and UACI are analyzed in the following sections.
2. In Table 6, compared with the method described in [10], the performance of the proposed method is somewhat low for entropy and NPCR. If possible, the reason can be given.
3. In the process of analyzing the noise attack, how to add the Gaussian noise to the cphertext should be given. Which kind of mode, i.e., an additive model or a multiplication model, is used?
Reviewer 2 Report
Comments and Suggestions for Authors
Abstract:
- what do the authors mean by "to promote the quality of the encrypted images"? Technically speaking this syntagma has no correct meaning.
- please avoid writing "and genetic algorithm" / "the genetic algorithm": there are many genetic algorithms...
- please rewrite "The whole cryptosystem is divided into three stages: firstly, the generation of a difference matrix; secondly, its utilization for initial encryption and the resulting initial ciphertext image already has a relatively good randomness; and finally, multiple rounds of local genetic operations to optimize the output."; it is confusing as a cryptosystem also includes the decryption function; the authors may have referred to the encryption function here (and first, parameter generation).
The security analysis is rather shallow, it needs more sophisticated technical additions. Please check thoroughly the corresponding literature. I would like to see a much more formal analysis of these aspects.
Please avoid using terms like "powerful cryptosystem", there is no such thing.
Comments on the Quality of English Language
Please use a grammar checker, spotted issues and typos.
